# Inhibitors of Endocannabinoids’ Enzymatic Degradation as a Potential Target of the Memory Disturbances in an Acute N-Methyl-D-Aspartate (NMDA) Receptor Hypofunction Model of Schizophrenia in Mice

**DOI:** 10.3390/ijms241411400

**Published:** 2023-07-13

**Authors:** Marta Kruk-Slomka, Bartlomiej Adamski, Tomasz Slomka, Grazyna Biala

**Affiliations:** 1Department of Pharmacology and Pharmacodynamics, Medical University of Lublin, 4a Chodzki Street, 20-093 Lublin, Poland; bartlomiejadamski1998@gmail.com (B.A.); grazyna.biala@umlub.pl (G.B.); 2Department of Medical Informatics and Statistics with E-Health Lab, Medical University of Lublin, Jaczewskiego 4 Street, 20-954 Lublin, Poland; tomasz.slomka@umlub.pl

**Keywords:** endocannabinoids, schizophrenia-like cognitive disorders, FAAH and MAGL inhibitor, NMDA receptor hypofunction, passive avoidance test, mice

## Abstract

Treating schizophrenia with the available pharmacotherapy is difficult. One possible strategy is focused on the modulation of the function of the endocannabinoid system (ECS). The ECS is comprised of cannabinoid (CB) receptors, endocannabinoids and enzymes responsible for the metabolism of endocannabinoids (fatty acid hydrolase (FAAH) and monoacylglycerol lipase (MAGL)). Here, the aim of the experiments was to evaluate the impact of inhibitors of endocannabinoids’ enzymatic degradation in the brain: KML-29 (MAGL inhibitor), JZL-195 (MAGL/FAAH inhibitor) and PF-3845 (FAAH inhibitor), on the memory disturbances typical for schizophrenia in an acute N-methyl-D-aspartate (NMDA) receptor hypofunction animal model of schizophrenia (i.e., injection of MK-801, an NMDA receptor antagonist). The memory-like responses were assessed in the passive avoidance (PA) test. A single administration of KML-29 or PF-3845 had a positive effect on the memory processes, but an acute administration of JZL-195 impaired cognition in mice in the PA test. Additionally, the combined administration of a PA-ineffective dose of KML-29 (5 mg/kg) or PF-3845 (3 mg/kg) attenuated the MK-801-induced cognitive impairment (0.6 mg/kg). Our results suggest that the indirect regulation of endocannabinoids’ concentration in the brain through the use of selected inhibitors may positively affect memory disorders, and thus increase the effectiveness of modern pharmacotherapy of schizophrenia.

## 1. Introduction

Schizophrenia is a chronic and very severe neuropsychiatric disorder that affects up to 1% of the human population. The clinical picture of this illness consists of three types of symptoms—positive (psychosis), negative (apathy) and cognitive symptoms [1]. The available pharmacotherapy of schizophrenia primarily focuses on the control of psychosis and has a limited efficacy on the negative symptoms. On the other hand, cognitive disorders in the course of schizophrenia are not subject to treatment with classic antipsychotic drugs, and additionally, sometimes such pharmacotherapy may exacerbate these disorders [2]. This is a very important medical problem because the large changes in cognitive functions are very often noticeable in people suffering from schizophrenia. It is estimated that 85% of patients struggling with schizophrenia experience cognitive issues with planning, memory and/or attention [3]. Therefore, there is a need to search for a new schizophrenia treatment strategy that would control not only positive and/or negative symptoms, but also affect the patient’s cognitive functions. Several lines of preclinical and clinical studies indicated clear correlations between the endocannabinoid system (ECS) and schizophrenia-like symptoms [4,5,6,7,8]. Thus, one possible strategy to control all symptoms of schizophrenia, including the cognitive impairment, is to modulate the function of the ECS.

The ECS consists of two types of cannabinoid (CB) receptors: CB1 and CB2, and their endogenous ligands, endocannabinoids. The two best-known endocannabinoids, anandamide (AEA) and 2-arachidynoglycerol (2-AG), are degraded by fatty acid hydrolase (FAAH) and monoacylglycerol lipase (MAGL), respectively. FAAH and MAGL enzymes are highly expressed in the central nervous system (CNS), as well as in the peripheral tissues. In the CNS, FAAH is primarily found in principal neurons of the hippocampus, cerebral cortex, cerebellum and olfactory bulb. This enzyme localizes on intracellular membranes in postsynaptic neurons [7]. In turn, MAGL exists in two tissue-specific splicing isoforms and is associated with membranes of presynaptic neurons. In the CNS, MAGL can be found in the hippocampus and cerebellum, as well as in the anterior thalamus [7]. 

The function of the ECS might be modulated in a direct way, through CB receptor ligands, or indirectly by FAAH and MAGL inhibitors. Modulation of ECS function via CB1 or CB2 receptor ligands has been widely described in our previous behavioral experiments [4,5,6], as well as in biochemical studies [9,10,11,12]. However, the data concerning the inhibition of endocannabinoids’ enzymatic degradation in the brain as a potential medical target in the treatment of many CNS disorders are very promising and still need to be enriched. In mice lacking the MAGL gene (MAGL knock-out mice), a substantial reduction in brain 2-AG hydrolytic activity can be observed, accompanied by ten-fold elevations in brain 2-AG levels and concomitant reduced AEA levels [12]. Among all MAGL inhibitors, the most pharmacological hope is associated with the two selective MAGL inhibitors: JZL-184 and its analog KML-29. These MAGL inhibitors are derived from the O-aryl carbamate class and show good selectivity (>100-fold) for MAGL over FAAH [13]. Furthermore, all potent inhibitors of MAGL activity described so far are claimed as irreversible because they produce their action by forming covalent bonds with reactive cysteine or serine residues. Similarly, the administration of both inhibitors caused a dramatic decrease in brain MAGL hydrolysis activity and concomitantly elevated 2-AG levels in the brain in mice. These metabolic and biochemical modifications are associated with several CB1- and/or CB2-dependent behavioral effects. JZL-184 elicited cannabinomimetic untoward effects, including hypothermia and hypomotility, while KML-29 did not have such effects [13,14]. In turn, in mice with the deletion of the FAAH gene (FAAH knock-out mice), and after injection of the selective FAAH inhibitors (e.g., PF-3845, PF-004457845, URB-597 or SSR411298), elevation of the level of AEA was observed. Literature data also described other behavioral responses in animals after administration of dual-inhibitors of FAAH and MAGL. It has been reported that FAAH/MAGL inhibitors (e.g., JZL-195) caused catalepsy that was not seen with selective FAAH or MAGL inhibitors [13,14]. There are also studies describing the negative impact of JZL-195 on sensory and motor processes [15]. Additionally, JZL-195 exhibited its pharmacological effect in reducing neuropathic and inflammatory pain, which has been demonstrated in several scientific publications [15,16,17]. JZL-195 also showed the most effective antinociceptive activity compared to selective MAGL or FAAH inhibitors [13]. It was noticed that JZL-195 had anti-allodynic activity, and the effect itself was much stronger than the anti-allodynic effect caused by selective inhibitors of only one enzyme [16,17]. Interestingly, JZL-195 turned out to be effective in reducing inflammation in the CNS due to its potential antioxidant properties, which may suggest the role of JZL-195 in the context of neuroprotective activity or neurodegeneration inhibition [18].

As stated, pharmacological inhibition of the activity of MAGL and/or FAAH can modulate many behavioral effects [9,10,11,12,13,14,15,16,17,18]. However, a new and still poorly researched strategy is the assessment of the FAAH and MAGL inhibitors on the course of schizophrenia symptoms. As mentioned before, the action of FAAH and/or MAGL inhibitors leads to an increase in the concentration of endocannabinoids in the CNS, and thus an increase in the effective ligand–receptor connections, which indirectly stimulates the functions of the ECS. Moreover, changes in the tissue concentrations of AEA and 2-AG have been observed in many neurological and psychiatric conditions, e.g., schizophrenia. Patients with acute paranoid schizophrenia have higher levels of AEA in the blood, plasma and cerebrospinal fluid, rather than a decrease in AEA during remission [9,10,11]. These results could suggest the participation of AEA in the regulation of homeostasis in patients suffering from schizophrenia. In addition, the change in the level of endocannabinoids can be helpful with the support of pharmacotherapy with neuroleptics [19]. 

The aim of this study was, for the first time, to assess and compare the impact of FAAH and/or MAGL inhibitors: KML-29 (a selective MAGL inhibitor), PF-3845 (a selective FAAH inhibitor) and JZL-195 (a dual MAGL and FAAH inhibitor), on the different stages of memory processes (acquisition, consolidation and retrieval) in mice, as well as evaluation of the effects of the above compounds in an experimental mouse model of schizophrenia. Based on one of the hypotheses of the pathology of schizophrenia, to induce the disorders typical for this disease, the N-methyl-D-aspartic acid (NMDA) receptor antagonist (MK-801) was used. The most well-known hypothesis of schizophrenia is associated with glutamatergic neurotransmission dysfunction. Pathogenesis of schizophrenia involves a diminished function or density of NMDA receptors caused by abnormalities in glutamate (Glu) neurotransmission. The potential relevance of Glu in the pathophysiology of schizophrenia was discovered via research with NMDA receptor antagonists, such as MK-801 [20]. This compound is a commonly accepted animal model of schizophrenia as it provokes a wide range of schizophrenia-like symptoms in rodents. In the present study, MK-801 was used to induce cognitive disorders in mice, which correspond to the cognitive symptoms observed in schizophrenia in humans. To assess memory-related effects in mice, we used the passive avoidance (PA) test, which is a commonly used behavioral test that allows evaluating different stages of memory depending on the drug treatment [20,21,22,23].

The obtained results will be used to broaden the knowledge about the involvement of the ECS in cognitive functions, through the indirect ECS modulation in memory processes, as well as the role of indirect ECS modulation in cognitive issues typical of schizophrenia. We hope that the conducted research could be used to initiate research at the clinical level and contribute to the emergence of new therapy directions in the treatment of schizophrenia with the use of selected inhibitors of enzymes that break down endocannabinoids in the brain.

## 2. Results

In the first step, for the first time, we estimated the influence of an acute administration of FAAH and/or MAGL inhibitors: KML-29, PF-2845 and JZL-195, on the different stages of long-term memory, i.e., acquisition, consolidation and retrieval, in the PA test in mice. The scheme of the behavioral experiments of the first step is shown in Figure 1.

### 2.1. The Influence of an Acute Injection of Selected Enzymatic Degradation of Endocannabinoids’ Inhibitors on the Long-Term Memory in Mice in the PA Test

#### 2.1.1. The Influence of an Acute Injection of MAGL Inhibitor KML-29 on the Long-Term Memory in Mice in the PA Test 

##### Acquisition of Memory

The one-way ANOVA revealed that administration of acute ip doses of KML-29 (1, 5, 20 and 40 mg/kg) had a statistically significant effect on the LI values for long-term memory acquisition (F(4,46) = 6.508; *p* = 0.0037). Moreover, the post hoc Tukey’s test indicated that KML-29, only at the highest dose (40 mg/kg), significantly increased the LI values in mice, compared to those in the vehicle-treated control group (*p* < 0.05) (Figure 2A). KML-29 as a MAGL inhibitor at this dose (40 mg/kg) had a positive influence on the long-term acquisition of memory and learning processes in the PA test in mice.

##### Consolidation of Memory

The one-way ANOVA further revealed that administration of acute ip doses of KML-29 (1, 5, 20 and 40 mg/kg) had a statistically significant effect on the LI values for long-term memory consolidation (F(4,37) = 4.044; *p* = 0.0081). Moreover, the post hoc Tukey’s test indicated that KML-29 at the two highest doses (20 and 40 mg/kg) significantly increased the LI values in mice, compared to those in the vehicle-treated control group (*p* < 0.05) (Figure 2B). KML-29 as a MAGL inhibitor at these doses (20 and 40 mg/kg) had a positive influence on the long-term consolidation of memory and learning processes in the PA test in mice.

##### Retrieval of Memory

Then, the one-way ANOVA revealed that administration of acute ip doses of KML-29 (1, 5, 20 and 40 mg/kg) had a statistically significant effect on the LI values for long-term memory retrieval (F(4,36) = 11.12; *p* < 0.0001). Moreover, the post hoc Tukey’s test indicated that KML-29 at the two highest doses (20 and 40 mg/kg) significantly increased the LI values in mice, compared to those in the vehicle-treated control group (*p* < 0.01 for the dose of 20 mg/kg and *p* < 0.001 for the dose of 40 mg/kg) (Figure 2C). KML-29 as a MAGL inhibitor at these doses (20 and 40 mg/kg) had positive influence on the long-term retrieval of memory and learning processes in the PA test in mice.

#### 2.1.2. The Influence of an Acute Injection of FAAH Inhibitor PF-3845 on the Long-Term Memory in Mice in the PA Test

##### Acquisition of Memory

For long-term memory acquisition, the one-way ANOVA revealed that administration of acute ip doses of PF-3845 (1, 3 and 10 mg/kg) had no statistically significant effect on the LI values (F(3,32) = 0.4966; *p* = 0.6875) (Figure 3A). PF-3845 as a FAAH inhibitor at this range of doses (1–10 mg/kg) had no influence on the long-term acquisition of memory and learning processes in the PA test in mice.

##### Consolidation of Memory

For long-term memory consolidation, the one-way ANOVA revealed that administration of acute ip doses of PF-3845 (1, 3 and 10 mg/kg) had no statistically significant effect on the LI values (F(3,33) = 0.1359; *p* = 0.9378) (Figure 3B). PF-3845 as a FAAH inhibitor at this range of doses (1–10 mg/kg) had no influence on the long-term consolidation of memory and learning processes in the PA test in mice.

##### Retrieval of Memory

The one-way ANOVA revealed that administration of acute ip doses of PF-3845 (1, 3 and 10 mg/kg) had a statistically significant effect on the LI values for long-term memory retrieval (F(3,32) = 3.655; *p* = 0.0238). Moreover, the post hoc Tukey’s test confirmed that PF-3845 at the highest dose (10 mg/kg) significantly increased the LI values in mice compared to those in the vehicle-treated control group (*p* < 0.05) (Figure 3C). PF-3845 as a FAAH inhibitor at this dose (10 mg/kg) had a positive influence on the long-term retrieval of memory and learning processes in the PA test in mice.

#### 2.1.3. The Influence of an Acute Injection of FAAH/MAGL Inhibitor JZL-195 on the Long-Term Memory in Mice in the PA Test

##### Acquisition of Memory

Here, the one-way ANOVA revealed that administration of acute ip doses of JZL-195 (5, 10 and 20 mg/kg) had a statistically significant effect on the LI values for long-term memory acquisition (F(3,37) = 8.065; *p* = 0.0003). Moreover, the post hoc Tukey’s test indicated that JZL-195 at the doses of 5 and 10 mg/kg significantly decreased the LI values in mice, compared to those in the vehicle-treated control group (*p* < 0.01 for the dose of 5 mg/kg and *p* < 0.001 for the dose of 10 mg/kg) (Figure 4A). JZL-195 as a FAAH/MAGL mixed inhibitor at these doses (5 and 10 mg/kg) had a negative influence on the long-term acquisition of memory and learning processes in the PA test in mice.

##### Consolidation of Memory

The one-way ANOVA further revealed that administration of acute ip doses of JZL-195 (5, 10 and 20 mg/kg) had a statistically significant effect on the LI values for long-term memory consolidation (F(3,33) = 3.695; *p* = 0.0225). Moreover, the post hoc Tukey’s test indicated that JZL-195 at the dose of 10 mg/kg significantly decreased the LI values in mice, compared to those in the vehicle-treated control group (*p* < 0.01) (Figure 4B). JZL-195 as a FAAH/MAGL mixed inhibitor at this dose (10 mg/kg) had a negative influence on the long-term consolidation of memory and learning processes in the PA test in mice.

##### Retrieval of Memory

The one-way ANOVA then revealed that administration of acute ip doses of JZL-195 (5, 10 and 20 mg/kg) had a statistically significant effect on the LI values for long-term memory retrieval (F(3,32) = 4.913; *p* = 0.0070). Moreover, the post hoc Tukey’s test indicated that JZL-195 at the doses of 5 and 10 mg/kg significantly decreased the LI values in mice, compared to those in the vehicle-treated control group (*p* < 0.05 for the dose of 5 mg/kg and *p* < 0.01 for the dose of 10 mg/kg) (Figure 4C). JZL-195 as a FAAH/MAGL mixed inhibitor at these doses (5 and 10 mg/kg) had a negative influence on the long-term retrieval of memory and learning processes in the PA test in mice.

In the next step of the experiments, we assessed, also for the first time, the impact of the tested inhibitors (KML-29, PF-3845, JZL-195) on the memory impairment provoked by an acute injection of MK-80 during the three stages (acquisition, consolidation and retrieval) of long-term memory in the PA test in mice. The scheme of the behavioral experiments of the second step is shown in Figure 5.

### 2.2. The Influence of an Acute Injection of Selected Inhibitors of Enzymatic Degradation of Endocannabinoids on MK-801-Induced Long-Term Memory Disturbances in Mice in the PA Test

Based on the results obtained from the above-described pilot experiments, the non-effective doses of the tested inhibitors (KML-29 (5 mg/kg), PF-3845 (3 mg/kg), JZL-195 (20 mg/kg)) were then chosen for the next behavioral experiments, evaluating their influence on the memory impairment, provoked by an acute injection of MK-801 (0.6 mg/kg), using the PA test in mice. The negative influence of MK-801 at the dose of 0.6 mg/kg on the long-term memory processes was confirmed in our previously described experiments [4,5,6,23].

#### 2.2.1. The Influence of KML-29 on the Memory Impairment Provoked by an Acute Administration of MK-801 in the PA Test in Mice

##### Acquisition of Memory

For long-term memory acquisition, two-way ANOVA analyses revealed that there was a statistically significant effect caused by KML-29 (5 mg/kg) pretreatment (F(1,31) = 6.650; *p* = 0.0149), as well as MK-801 (0.6 mg/kg) treatment (F(1,31) = 23.13; *p* < 0.0001), and there was also a statistically significant effect caused by the interactions (F(1,31) = 4.919; *p* = 0.0340).

The post hoc Bonferroni’s test confirmed that MK-801 at the dose of 0.6 mg/kg significantly decreased the LI values in mice in the PA test, in comparison to the vehicle/vehicle-treated mice, pointing to the amnestic effect of this drug (*p* < 0.001; Bonferroni’s test). Moreover, an acute injection of KML-29 (5 mg/kg) attenuated the amnestic effect of MK-801 (0.6 mg/kg) (*p* < 0.05; Bonferroni’s test) in the PA test in mice (Figure 6A).

##### Consolidation of Memory

For long-term memory consolidation, two-way ANOVA analyses revealed that there was a statistically significant effect caused by KML-29 (5 mg/kg) pretreatment (F(1,32) = 12.83; *p* = 0.0011), as well as MK-801 (0.6 mg/kg) treatment (F(1,32) = 23.94; *p* < 0.0001), but there was no statistically significant effect caused by the interactions (F(1,32) = 3.191; *p* = 0.0835).

In turn, the post hoc Bonferroni’s test confirmed that MK-801 at the dose of 0.6 mg/kg significantly decreased the LI values in mice in the PA test, in comparison to the vehicle/vehicle-treated mice, pointing to the amnestic effect of this drug (*p* < 0.001; Bonferroni’s test). Moreover, an acute injection of KML-29 (5 mg/kg) attenuated the amnestic effect of MK-801 (0.6 mg/kg) (*p* < 0.01; Bonferroni’s test) in the PA test in mice (Figure 6B).

##### Retrieval of Memory

For long-term memory retrieval, two-way ANOVA analyses revealed that there was a statistically significant effect caused by KML-29 (5 mg/kg) pretreatment (F(1,31) = 8.261; *p* = 0.0073), as well as MK-801 (0.6 mg/kg) treatment (F(1,31) = 10.68; *p* = 0.0027), and there was also a statistically significant effect caused by the interactions (F(1,31) = 6.547; *p* = 0.0156).

In turn, the post hoc Bonferroni’s test confirmed that MK-801 at the dose of 0.6 mg/kg significantly decreased the LI values in mice in the PA test, in comparison to the vehicle/vehicle-treated mice, pointing to the amnestic effect of this drug (*p* < 0.01; Bonferroni’s test). Moreover, an acute injection of KML-29 (5 mg/kg) attenuated the amnestic effect of MK-801 (0.6 mg/kg) (*p* < 0.01; Bonferroni’s test) in the PA test in mice (Figure 6C).

#### 2.2.2. The Influence of PF-3845 on the Memory Impairment Provoked by an Acute Administration of MK-801 in the PA Test in Mice

##### Acquisition of Memory

For long-term memory acquisition, two-way ANOVA analyses revealed that there was a statistically significant effect caused by PF-3845 (3 mg/kg) pretreatment (F(1,31) = 10.49; *p* = 0.00029), as well as MK-801 (0.6 mg/kg) treatment (F(1,31) = 23.11; *p* < 0.0001), but there was no statistically significant effect caused by the interactions (F(1,31) = 3.646; *p* = 0.0655).

However, the post hoc Bonferroni’s test indicated that MK-801 at the dose of 0.6 mg/kg significantly decreased the LI values in mice in the PA test, in comparison to the vehicle/vehicle-treated mice, pointing to the amnestic effect of this drug (*p* < 0.001; Bonferroni’s test). Moreover, an acute injection of PF-3845 (3 mg/kg) attenuated the amnestic effect of MK-801 (0.6 mg/kg) (*p* < 0.01; Bonferroni’s test) in the PA test in mice (Figure 7A).

##### Consolidation of Memory

For long-term memory consolidation, two-way ANOVA analyses revealed that there was a statistically significant effect caused by PF-3845 (3 mg/kg) pretreatment (F(1,31) = 10.31; *p* = 0.0031), as well as MK-801 (0.6 mg/kg) treatment (F(1,31) = 35.61; *p* < 0.0001), but there was no statistically significant effect caused by the interactions (F(1,31) = 2.916; *p* = 0.0977).

In turn, the post hoc Bonferroni’s test indicated that MK-801 at the dose of 0.6 mg/kg significantly decreased the LI values in mice in the PA test, in comparison to the vehicle/vehicle-treated mice, pointing to the amnestic effect of this drug (*p* < 0.001; Bonferroni’s test). Moreover, an acute injection of PF-3845 (3 mg/kg) attenuated the amnestic effect of MK-801 (0.6 mg/kg) (*p* < 0.01; Bonferroni’s test) in the PA test in mice (Figure 7B).

##### Retrieval of Memory

For long-term memory retrieval, two-way ANOVA analyses revealed that there was a statistically significant effect caused by PF-3845 (3 mg/kg) pretreatment (F(1,31) = 21.35; *p* < 0.0001), as well as MK-801 (0.6 mg/kg) treatment (F(1,31) = 56.30; *p* < 0.0001), and there was also a statistically significant effect caused by the interactions (F(1,31) = 19.10; *p* < 0.0001).

The post hoc Bonferroni’s test confirmed that MK-801 at the dose of 0.6 mg/kg significantly decreased the LI values in mice in the PA test, in comparison to the vehicle/vehicle-treated mice, pointing to the amnestic effect of this drug (*p* < 0.001; Bonferroni’s test). Moreover, an acute injection of PF-3845 (3 mg/kg) attenuated the amnestic effect of MK-801 (0.6 mg/kg) (*p* < 0.001; Bonferroni’s test) in the PA test in mice (Figure 7C).

#### 2.2.3. The Influence of JZL-195 on the Memory Impairment Provoked by an Acute Administration of MK-801 in the PA Test in Mice

##### Acquisition of Memory

For long-term memory acquisition, two-way ANOVA analyses revealed that there was a statistically significant effect caused by MK-801 (0.6 mg/kg) treatment (F(1,31) = 65.94; *p* < 0.0001), but there was no statistically significant effect caused by JZL-195 (20 mg/kg) pretreatment (F(1,31) = 0.03635; *p* = 0.8500), and there was no statistically significant effect caused by the interactions (F(1,31) = 0.07111; *p* = 0.7915).

The post hoc Bonferroni’s test confirmed that MK-801 at the dose of 0.6 mg/kg significantly decreased the LI values in mice in the PA test, in comparison to the vehicle/vehicle-treated mice, pointing to the amnestic effect of this drug (*p* < 0.001; Bonferroni’s test). An acute injection of JZL-195 (20 mg/kg) had no influence on the amnestic effect of MK-801 (0.6 mg/kg) in the PA test in mice (Figure 8A).

##### Consolidation of Memory

For long-term memory consolidation, two-way ANOVA analyses revealed that there was a statistically significant effect caused by MK-801 (0.6 mg/kg) treatment (F(1,33) = 58.69; *p* < 0.0001), but there was no statistically significant effect caused by JZL-195 (20 mg/kg) pretreatment (F(1,33) = 0.8949; *p* = 0.3510), and there was no statistically significant effect caused by the interactions (F(1,33) = 1.361; *p* = 0.2517).

The post hoc Bonferroni’s test confirmed that MK-801 at the dose of 0.6 mg/kg significantly decreased the LI values in mice in the PA test, in comparison to the vehicle/vehicle-treated mice, pointing to the amnestic effect of this drug (*p* < 0.001; Bonferroni’s test). An acute injection of JZL-195 (20 mg/kg) had no influence on the amnestic effect of MK-801 (0.6 mg/kg) in the PA test in mice (Figure 8B).

##### Retrieval of Memory

For long-term memory retrieval, two-way ANOVA analyses revealed that there was a statistically significant effect caused by MK-801 (0.6 mg/kg) treatment (F(1,33) = 60.58; *p* < 0.0001), but there was no statistically significant effect caused by JZL-195 (20 mg/kg) pretreatment (F(1,33) = 1.195; *p* = 0.2821), and there was no statistically significant effect caused by the interactions (F(1,33) = 1.886; *p* = 0.1789).

The post hoc Bonferroni’s test confirmed that MK-801 at the dose of 0.6 mg/kg significantly decreased the LI values in mice in the PA test, in comparison to the vehicle/vehicle-treated mice, pointing to the amnestic effect of this drug (*p* < 0.001; Bonferroni’s test). An acute injection of JZL-195 (20 mg/kg) had no influence on the amnestic effect of MK-801 (0.6 mg/kg) in the PA test in mice (Figure 8C).

## 3. Discussion

Schizophrenia is one of the most severe mental disorders and is characterized by complex issues, including psychosis and/or apathy and/or cognitive issues. The current pharmacotherapy of schizophrenia, using antipsychotic drugs (neuroleptics), is focused on limiting psychosis, and has no influence on the negative and cognitive symptoms. Therefore, a novel treatment option for all symptoms of schizophrenia is needed. In this context, an interesting goal seems to be the ECS and modulation of its functions due to interconnections between the ECS and schizophrenia-like symptoms [4,5,6,9,10,11,19,24]. Thus, changes in endocannabinoids’ levels have also been shown in patients suffering from different types of schizophrenia [9,10,19]. However, there is still a distinct lack of evidence regarding the influence of inhibitors of enzymatic degradation of endocannabinoids on the schizophrenia-like responses, especially cognition symptoms.

Based on the literature data mentioned above [4,5,6,9,10,11,19,24], using an animal model of schizophrenia, in the current study, we aimed to investigate this issue by examining how indirect alteration of endocannabinoid levels affects the memory-related responses. The aim of the study was to determine the influence of the selected inhibitors of enzymatic degradation of endocannabinoids on the different stages of memory processes (acquisition, consolidation and retrieval), as well as on the memory-related disorders in the context of schizophrenia. To assess the cognitive functions in mice, we used the PA test, commonly used in pharmacological studies. Among all modulators of enzyme-metabolizing endocannabinoids, in the present experiments, we used three new compounds: KML-29 (a selective MAGL inhibitor), PF-3845 (a selective FAAH inhibitor) and JZL-195 (a dual MAGL and FAAH inhibitor). MK-801 was used to obtain cognitive-related disorders in mice, typical for schizophrenia symptoms. In the first step of the study, the influence of an acute administration of the inhibitors (KML-29, PF-3845 and JZL-195) on the processes of memory acquisition, consolidation and retrieval in the PA test in mice was assessed. In the second step, in order to determine the role of the ECS in schizophrenia, the influence of an acute administration of the above-mentioned inhibitors on the MK-801-provoked amnestic effects in mice was investigated.

Our studies revealed that an acute administration of the MAGL inhibitor KML-29 resulted in a significant increase in the value of the LI parameter in the memory acquisition phase (the dose of 40 mg/kg), in the memory consolidation phase (the doses of 20 and 40 mg/kg), as well as in the memory retrieval phase (the doses of 20 and 40 mg/kg), which could prove the positive effect of this inhibitor on the different memory stages in mice evaluated in the PA test. The lower doses of KML-29 (1 or 5 mg/kg) had no statistically significant influence on the LI value. The presented results are in line with our previous data, which revealed that an acute injection of another MAGL inhibitor, JZL-184 (4 mg/kg), improved the acquisition of memory and learning processes in the PA test in mice [6]. Similarly, Ratano et al. [25] showed an improvement in cognitive functions in the PA test in rats during the memory consolidation stage, also using JZL-184 (0.5 mg/kg). It could be suggested that the memory improvement induced by the JZL-184 was not strictly related to the CB1 receptor mechanism, because in the next study, the same authors revealed that the acute administration of SR141716 (0.3 mg/kg), a CB1 receptor antagonist, did not reverse the memory improvement caused by JZL-184 (0.5 mg/kg) in rats in the PA test. However, the reversal occurred in the next stage of the research using the CB2 receptor antagonist, SR144528 (0.03 mg/kg), which reversed the memory improvement provoked by JZL-184 (0.5 mg/kg) in the memory consolidation stage in the PA test in rats [25]. The experiment showed that the pro-cognitive effect of JZL-184 could be associated with the involvement of the CB2 receptor. Thus, it could suggest that the inhibition of MAGL after an acute injection of different inhibitors (e.g., JZl-184 or KML-29) causes pharmacological enhancement of the 2-AG level, facilitating memory-related processes through activation of CB2 receptor signaling. Similar results were obtained by Aymerich et al. [26], who revealed that the neuroprotective effect of JZL-184 was reversed by a CB2 receptor antagonist, AM-630. The CB1 receptor antagonist, rimonabant, had no influence on the JZL-184-provoked neuroprotective effects. Additionally, an acute administration of the CB2 receptor agonist mimicked the effects of JZL-184 [26], suggesting that the pharmacological properties caused by JZL-184 are similar to those of the CB2 receptor agonist. However, it should be noted that MAGL inhibitors show a similar, but somewhat broader spectrum of CB1-dependent behavioral effects, and at high doses, can lead to desensitization and downregulation of CB1 receptors. It has been found that, at the higher dose, JZL-184 was able to impair memory processes [27]. Thus, JZL-184 may also act through the CB1 receptors. Pharmacological blockade of MAGL has been reported to result in dramatic elevation in brain 2-AG levels, suggesting that the selective increase in brain 2-AG levels is associated with memory disruption. Additionally, it has been reported that oral administration of another MAGL inhibitor—SAR127303—causes spatial, episodic and working memory impairment in rats using the novel object tests, the Y maze and the Morris water maze tests, respectively. In addition, it was noted that the administration of the CB1 receptor antagonist—rimonabant—reduces the amnestic effect of SAR127303; hence, it can be suggested that the indirect activation of CB1 receptors by the described endocannabinoid breakdown inhibitor is responsible for the deterioration of cognitive functions [27,28]. In the context of the influence on the memory-related responses by inhibiting MAGL, other research seems to be important and controversial. Chen et al. [29] showed the improvement in spatial learning and memory in a mouse model of Alzheimer’s disease (AD) after using the MAGL inhibitor (JZL-184), assessed in the Morris water maze test. An additional study revealed that the use of JZL-184 at different time intervals and different dosing schedules led to a decrease in the concentration of beta amyloid in the cells of the cerebral cortex and hippocampus, which correlates with the reduction of neurodegeneration observed in patients with AD. Moreover, there was a noticeable decrease in the expression of beta-secretase 1, which is one of the most important enzymes involved in the synthesis of pathological beta amyloid. It was also revealed that JZL-184 led to a reduction in inflammation in the brain by reducing the amounts of reactive astrocytes, which promotes the potential inhibition of AD progression [29]. This anti-inflammatory property can be taken into consideration also in the context of schizophrenia symptoms, i.e., cognitive ones [18,29]. Therefore, taking the above into account, the pro-cognitive mechanism of the MAGL inhibitor KML-29 observed in our studies might be related to the indirect activation of CB2 subtype receptors. This mechanism of action is probably crucial for the reversal of the memory impairment provoked by MK-801 by KML-29 obtained in our studies.

With regards to the second inhibitor, our results revealed that an acute administration of the FAAH inhibitor PF-3845 significantly increased the value of the LI parameter only in the memory retrieval stage in the PA test in mice at the dose of 10 mg/kg, which could prove the positive effect of the tested compound on the memory retrieval process. There was no statistically significant effect of an acute administration of PF-3845 in the dose range of 1 to 10 mg/kg on the acquisition and consolidation processes in the PA test in mice. We have found similar literature data that are in accordance with our results, in which PF-3845, as a FAAH inhibitor, improved the three stages of memory and learning processes in the PA test in mice. One of the studies consistent with the above results is the study by Tchantchou et al. [30] on a mouse experimental model of traumatic brain injury. The study used male C57BL/6 mice that were traumatically injured during an operation that involved craniotomy and then firing an electrical impulse on the open brain, which caused brain injury. The mice were injected with PF-3845 in doses of 2, 5 and 10 mg/kg for various periods of therapy—from 3 to 14 days. A memory-related hippocampal response was assessed by Y-maze on the 10th day after the operation causing brain injuries. The PF-3845 inhibitor has been shown to have a positive effect on cognitive functions and working memory, and the administration of PF-3845 also significantly reduced the volume of brain neurodegeneration in the cortex and dentate gyrus at a dose of 5 mg/kg. Of interest, it was also noted that the overexpression of amyloid precursor protein (APP), typical for neurodegenerative disorders such as AD, decreased after administration of the PF-3845 inhibitor, and the expression of synaptophysin increased, which correlates with the potential positive effect on inhibiting or controlling the development of AD [30], further proving the neuroprotective effects of this compound. In addition, CB1 (AM-281) and CB2 (AM-630) receptor antagonists were used to explain the mechanisms of action of PF-3845. It was noted that the use of CB1 receptor antagonists significantly reversed the positive effects of PF-3845 on the memory processes, and the use of a CB2 receptor antagonist significantly but only partially reversed the effects of PF-3845, suggesting that the CB1 receptor plays a more crucial role in the pro-cognitive effect of PF-3845. In turn, the administration of both the CB1 and CB2 receptor inhibitors reversed the PF-3845 reduction in neurodegeneration processes [30]; thus, both CB receptor types can be involved. Moreover, the above results are partially consistent with studies describing the positive influence of other FAAH inhibitors on the memory-related processes. Among them, URB-597 has been studied most intensively. However, the influence of this compound on the memory processes is controversial. Inhibition or genetic deletion of FAAH, which substantially increases endogenous levels of AEA, has been found to enhance memory in rodents trained with procedures involving aversively motivated behavior (i.e., the Morris water maze test) or the PA test, with a context associated with foot shock [31,32,33]. Moreover, URB-597 (0.3 and 1 mg/kg) treatment could alleviate the negative influence of WIN-55,212-2, a partial CB1 receptor agonist, on cognition and memory [34,35]. These results indicate a potential of URB-597 to protect against memory deficits induced by cannabinoids. In turn, other memory-related studies have mostly shown impairment rather than enhancement after treatment with this FAAH inhibitor [28,34]. It appears that aversively motivated learning is most sensitive to be enhanced by FAAH manipulations, possibly due to the effects of FAAH inhibition on anxiety-related responses or coping behavior [35,36,37]. It should be noted here that an anxiolytic effect was also demonstrated after the administration of PF-3845 in the zero-maze test in mice; however, the administration of the CB1 and CB2 receptor antagonists did not significantly reverse this pharmacological effect [30]. It has been known that the administration of FAAH enzyme inhibitors may result in memory improvement, among others, through the CB receptor mechanism (by both CB1 and CB2 receptors). It can, therefore, be suggested that direct agonism of the CB1 receptor (using CB1 receptor ligands) leads to cognitive dysfunction, while indirect modulation of the activity of this receptor (using FAAH or MAGL enzyme inhibitors) does not lead to cognitive impairment [11]. These discrepancies may all be caused by a number of factors, including the choice of experimental methods as well as of the FAAH inhibitor. For example, it has been noticed that the use of FAAH inhibitors enhances memory in tests examining the functioning of memory using the memory of unpleasant stimuli, i.e., the mechanism of aversion [38].

Finally, we revealed that an acute administration of a dual MAGL/FAAH inhibitor—JZL-195—at a narrow range of doses (5–10 mg/kg) resulted in a statistically significant decrease in the value of the LI parameter during the acquisition, consolidation and retrieval memory stages in the PA test in mice. The presented results may suggest a negative impact of this compound on all memory phases assessed in mice in the PA test. The above action of the dual FAAH/MAGL inhibitor is not quite consistent with the available literature data. The latest data from the scientific work of Bajaj et al. [18] described that JZL-195 did not significantly affect cognitive impairment, but also did not lead to statistically significant deterioration of memory processes in mice in memory tests [18]. On the other hand, the literature data from Wise et al. [39] or Seillier et al. [40] may indicate a negative impact of JZL-195 on various CNS-related behaviors, including motor behavior or cognitive processes. The administration of JZL-195 resulted in a decrease in spatial memory in mice assessed in the Morris water maze test [39]. Moreover, administration of JZL-195 led to a reduction in the level of beta amyloid in the mouse hippocampus via the blood–brain barrier, as well as a significant increase in the activity of acetylcholinesterase and butyrylcholinesterase enzymes was also observed, which seems to be helpful in the treatment of AD [18,40] and can point to another interesting non-ECS-related mechanism of its cognitive effects.

In the second part of our experiments, we evaluated the influence of the FAAH and/or MAGL inhibitors described above on the schizophrenia-like cognitive disturbances in mice, provoked by a NMDA receptor antagonist, MK-801. We revealed that an acute injection of KML-29 (at the non-effective dose in the PA test—5 mg/kg) reversed the amnestic effect induced by the MK-801 injection (0.6 mg/kg) in all three stages of memory in mice assessed in the PA test. Additionally, an acute injection of PF-3845 (at the non-effective dose in the PA test—3 mg/kg) reversed the amnestic effect induced by the MK-801 injection (0.6 mg/kg) in all three stages of memory in mice assessed in the PA test. In turn, an acute injection of JZL-195 (at the non-effective dose in the PA test—20 mg/kg) had no influence on the memory impairment induced by MK-801 (0.6 mg/kg) in any of the stages of memory in mice in the PA test. In the context of the results presented in our manuscript associated with cognitive-related schizophrenia symptoms, the few literature data available describe the role of other FAAH or MAGL inhibitors in the control/modulation of schizophrenia symptoms in experimental animal models.

Available literature data provided results on the effect of the FAAH inhibitor URB-597 on memory disorders caused by dysfunction of the glutamatergic system. In a study by Seillier et al. [41], administration of URB-597 reversed the memory impairment induced by the NMDA receptor antagonist, phencyclidine (PCP). In our previous study [6], we also observed that an acute URB-597 administration in mice attenuated memory disorders caused by MK-801 [6], which is recognized as a pharmacological model of schizophrenia. In another study, Rivera et al. [42] showed that URB-597 (0.3 mg/kg) improved memory damage via administration of ethanol. The authors proved that URB-597 affects not only the functions of neurons, but also the element of the immune system in the hippocampus. URB-597 can modulate hippocampal microglial activation as well as recruitment, and that may be associated with improved hippocampal-dependent memory despite ethanol exposure [43]. These results are in accordance with our findings, where we observed that an administration of KML-29, a FAAH inhibitor, attenuated MK-801 (0.6 mg/kg)-induced memory impairment. In turn, a single injection of another MGL inhibitor, JZL-184, intensified the memory impairment induced by MK-801, or did not cause significant changes in cognitive disorders [6]. In our study, we revealed that PF-3845 also reversed the amnesia provoked by MK-801 in mice. However, the dual FAAH/MAGL inhibitor, JZL-195, had no influence on the memory disturbances induced by MK-801 administration in mice.

The above-described diverse effects of the FAAH and/or MAGL inhibitors on the MK-801-induced memory disorders in mice can firstly be related to their influence on the concentration of endocannabinoids, i.e., AEA and 2-AG, in the brain and the aforementioned activation of CB receptors (CB1 and/or CB2) [41]. Considering the action of endocannabinoid-degrading inhibitors on the dysfunction of the glutamatergic system induced by MK-801, attention should first be paid to the neurobiological effect at the basis of these interactions. In a glutamatergic model of schizophrenia, administration of an NMDA receptor antagonist has been shown to increase endocannabinoid release in the prefrontal cortex (PFC), nucleus accumbens (NAC) and ventral tegmental area (VTA) in rats, closely correlated with symptoms of schizophrenia. In addition, increasing the concentration of endocannabinoids may also enhance dopamine (DA) levels [41,42,43]. To confirm this statement, the elevated levels of endocannabinoids have been observed in patients during a psychotic episode. It is, therefore, believed that the increase in endogenous cannabinoid release is an effective compensation and consequence of excessive excitability of dopaminergic neurons, and could be evidence of the neuroprotective effect of endogenous cannabinoids [44]. These mechanisms may be responsible for the positive effects of FAAH or MAGL inhibitors on the MK-801-provoked memory disorders obtained in our study. However, it should be noted that the described effect of endocannabinoid-degrading inhibitors, i.e., increasing the concentration of endocannabinoids in the human body in the context of treating schizophrenic disorders, is not consistent with the work of Potvin et al. [45]. In this research, the authors noted that in the course of schizophrenia, there is a noticeable increase in the concentration of AEA in the cerebrospinal fluid, which they considered correlated with the occurrence of schizophrenia [45]. Interestingly, Joaquim et al. [46] noticed that during the prodromal periods of the development of schizophrenia, the level of endocannabinoids in the brain decreased [46], while elevated concentrations of endocannabinoids—2-AG and AEA—occurred only in the active stage of disease with psychoses present. Even more interestingly, endocannabinoid levels normalized in patients with schizophrenia during periods of remission [45]. Furthermore, the studies conducted on animal models of schizophrenia using the NMDA receptor antagonist, PCP, as well as post-mortem analysis of brain tissue of patients suffering from schizophrenia, showed a significant increase in the concentration of 2-AG in the PFC [46,47]. In an animal glutamatergic model of schizophrenia, NMDA receptors are blocked, which results in impaired cognitive processes and increased levels of 2-AG in certain areas of the brain. The increase in 2-AG due to hypofunction of the NMDA receptor may be an adaptive mechanism in this case [47,48].

In summary, the FAAH and/or MAGL inhibitors may have a different influence on the memory-related processes in mice in the PA test. The selective inhibitors (KML-29 or PF-3845) had a positive effect on the memory and learning, as well as on the memory disorders occurring in schizophrenia. On the other hand, the dual-inhibitors (e.g., JZL-195) had negative or no influence on the cognition (Figure 9).

Of course, it should also be noted that all the effects of FAAH and/or MAGL inhibitors observed in our experiments could often depend on many factors. Thus, the experiments themselves are limited for many reasons, e.g., behavioral tasks used, handling procedures, time of drug administration, as well as the kind of inhibitors and their selectivity. In our experiments, in order to assess and understand the memory-related effects, we used the PA test, which is commonly used to investigate emotional learning and memory processes in rodents. Depending on the procedure used, the PA test allows examining different memory durations (short-term and long-term memory), as well as different memory stages (acquisition, consolidation and retrieval). Usually, regarding the PA paradigm, response latency alterations have been thought to reflect the degree of memory; however, the emotionality (fear and/or anxiety) of animals can presumably affect the avoidance behavior. Another limitation that may have an influence on the memory-related responses observed in our experiments is associated with the selectivity of the inhibitors used, and thus their mechanisms of action—the influence of the AEA and/or 2-AG level in the brain. In our studies, we used three inhibitors (KML-29, PF-3845 and JZL-195) with different selectivity. They were used for a declared target (memory-related responses) that may suggest that FAAH and/or MAGL inhibition might be related to a wide spectrum of therapeutic actions in cognitive-related disorders. Naturally, we cannot rule out their impact on other animal behavior/disturbances. However, further studies are necessary to identify the clear mechanisms underlying the action of these inhibitors that may be used as pharmacological tools to independently manipulate AEA and 2-AG signaling, and to study their possible interactions.

## 4. Materials and Methods

### 4.1. Animals

The experiments were carried out on one-month-old naive male Swiss mice (Experimental Medicine Center, Lublin, Poland) weighing 20–30 g. Mice were housed in groups of 10 mice/home cage (38 × 22 × 18 cm), made of white plexiglass. The animals were maintained under standard laboratory conditions (12 h light/dark cycle, room temperature at 21 ± 1 °C, 55% humidity conditions) with free access to tap water and laboratory feeding (Agropol, Motycz, Poland) in their home cages, and adapted to the laboratory conditions for at least 1 week. Each experimental group consisted of 8 to 10 animals. All behavioral experiments were performed between 8:00 and 15:00.

### 4.2. Ethics

All experiments were carried out according to the permission of the Local Ethical Committee (Local Ethical Committee for Animal Experiments in Lublin: Approval Code: 3/2020; Approval Date: 24 February 2020).

All behavioral experiments were conducted according to the National Institute of Health Guidelines for the Care and Use of Laboratory Animals and the European Community Council Directive for the Care and Use of Laboratory Animals of 22 September 2010 (2010/63/EU). The authors complied with the ARRIVE guidelines to improve the reporting of animal research and improve the quality of the studies.

### 4.3. Drugs

The compounds tested were:

Inhibitors of enzymatic degradation of endocannabinoids:

KML-29 (highly potent and selective MAGL inhibitor; 4-[Bis(1,3-benzodioxol-5-yl)hydroxymethyl]-1-piperidinecarboxylic acid 2,2,2-trifluoro-1-(trifluoromethyl)ethyl ester; obtained from: Tocris Bioscience a Bio-Techne Brand, Biotechne, Warsaw, Poland). The doses used: 1, 5, 20, 40 mg/kg.

PF-3845 (selective FAAH inhibitor; N-3-Pyridinyl-4-[[3-[[5-(trifluoromethyl)-2-pyridinyl]oxy]phenyl]methyl]-1-piperidinecarboxamide; obtained from: Tocris Bioscience a Bio-Techne Brand, Biotechne, Warsaw, Poland). The doses used: 1, 3, 10 mg/kg.

JZL-195 (dual FAAH and MAGL inhibitor; 4-[(3-Phenoxyphenyl)methyl]-1-piperazinecarboxylic acid 4-nitrophenyl ester; obtained from: Tocris Bioscience a Bio-Techne Brand, Biotechne, Warsaw, Poland). The doses used: 5, 10, 20 mg/kg.

Pharmacological animal model of schizophrenia:

MK-801 (NMDA receptor antagonist; 5S-10,11-dihydro-5-metylo-5H-dibenzo[a,d]cyklohepten-5,10-imin; obtained from: Tocris Bioscience a Bio-Techne Brand, Biotechne, Warsaw, Poland). The dose used: 0.6 mg/kg.

All compounds were suspended in a 1% solution of Tween 80 (Sigma Aldrich, St. Louis, MO, USA) in saline (0.9% NaCl) and administered intraperitoneally (ip) at a volume of 10 mL/kg. Fresh drug solutions were prepared on each day of experimentation. Control groups received injections of saline with Tween 80 (vehicle) at the same volume and via the same route of administration.

### 4.4. Experimental Procedure

We used MK-801 as a pharmacological animal model of schizophrenia. An acute administration of MK-801 provoked schizophrenia-like symptoms in mice and manifested inter alia cognitive disturbances (correlation with the cognitive symptoms in humans). This procedure is commonly accepted [20] and was confirmed in our previous experiments [4,5,6,23]. Thus, the dose of MK-801 (0.6 mg/kg) that provoked memory impairment was selected from our previous experiments [4,5,6,23]. In turn, experimental doses of the inhibitors were selected based on the literature data [11,14,16,27]. In the present experiments, we evaluated, for the first time, the influence of an acute administration of inhibitors of enzymatic degradation of endocannabinoids (KML-29, PF-3845, JZL-195) on the memory and learning processes and on the above-described schizophrenia-like amnestic effects in mice, provoked by MK-801.

To assess and understand the memory-related effects, we used the PA test. The PA test is commonly used to investigate learning and memory processes in rodents [4,5,6,21,22,23].

PA apparatus: The apparatus of the PA test (obtained from ATANER, Lublin, Poland) consisted of a two-compartment acrylic box with a lightened compartment (10 × 13 × 15 cm) and darkened compartment (25 × 20 × 15 cm). The light chamber was illuminated by a fluorescent light (8 W) and was connected to the dark chamber, which was equipped with an electric grid floor. Entrance of animals to the dark box was punished by an electric foot shock (0.2 mA for 2 s).

PA experimental procedures: On the first day of training, mice were individually placed into the light compartment and allowed to explore the light box. After 30 s (habituation period), the guillotine door was raised to allow the mice to enter the dark compartment (pre-test). When the mice entered the dark compartment, the guillotine door was closed and an electric foot-shock (0.2 mA) of 2 s duration was immediately delivered to the animals via the grid floor. The latency time for entering the dark compartment was recorded (TL1). If the mouse failed to enter the dark box within 300 s, it was placed into the dark box, the door was closed, and then the electric foot-shock was delivered to the animal. In this case, the TL1 value was recorded as 300 s. In the subsequent trial, the same mice were again individually placed in the light compartment of the PA apparatus. After a 30 s adaptation period in the light (safe) chamber, the door between the compartments was raised and the time taken to re-enter the dark compartment was recorded (TL2) (test). No foot-shock was applied in this trial. If the animal did not enter the dark compartment within 300 s, the test was stopped and TL2 was recorded as 300 s.

For the memory-related responses, the changes in PA performance were expressed as the difference between retention and training latencies and were taken as a latency index (LI). The LI was calculated for each animal and reported as the ratio:LI = TL2 − TL1/TL1

TL1—the time taken to enter the dark compartment during the pre-test.

TL2—the time taken to re-enter the dark compartment during the test.

Depending on the procedure used, the PA test allows examining different durations of memory (short-term and long-term memory) according to the period between training and the test, as well as different stages of memory (acquisition, consolidation, retrieval) according to the time of the drug treatment. When mice were tested 24 h after TL1 the long-term fear memory was assessed. Drug administration before the first trial (before pre-test) should interfere with the acquisition of information, drug administration immediately after the first trial (after pre-test) should exert an effect on the process of consolidation, while the administration of tested compounds before the second trial (before test) should interfere with the retrieval of memory information [21,22,23].

### 4.5. Treatment

First, we estimated the impact of the selected inhibitors on the different stages of long-term memory in mice, using the PA test.

Inhibitors of enzymatic degradation of endocannabinoids: KML-29 (1, 5, 20, 40 mg/kg), PF-3845 (1, 3, 10 mg/kg) and JZL-195 (5, 10, 20 mg/kg), or vehicle for the control group, were administered ip 30 min before the first trial (acquisition of memory) (Table 1A) or immediately after the first trial (consolidation of memory) (Table 2A), and re-tested after 24 h. In the case of retrieval of memory, first, the mice were tested during the first trial. Inhibitors or vehicle for the control group were injected ip 30 min before retrieval, and the retrieval was carried out 24 h after the first trial (Table 1C).

Next, based on this pilot experiment, we chose the non-effective doses of inhibitors and evaluated the influence of these compounds on the memory-related disorders induced by MK-801 (0.6 mg/kg) in the PA test in mice.

Non-effective doses of the tested inhibitors: KML-29 (5 mg/kg), PF-3845 (3 mg/kg) and JZL-195 (20 mg/kg), or vehicle for the control group, were acutely administered ip, 15 min before an acute ip injection of MK-801 (0.6 mg/kg) or vehicle. Then, 15 min after the last injection, the mice were tested in the PA test during the first trial and re-tested 24 h later, for the assessment of long-term memory acquisition (Table 2A). In the case of consolidation of memory, non-effective doses of the tested inhibitors: KML-29 (5 mg/kg), PF-3845 (3 mg/kg) and JZL-195 (20 mg/kg), or vehicle for the control group, were acutely administered ip, 15 min before an acute ip injection of MK-801 (0.6 mg/kg) or vehicle, immediately after the first trial. Then, the mice were re-tested 24 h later (Table 2B). In the case of retrieval of memory, the mice were tested during the first trial. Then, 24 h later, non-effective doses of the tested inhibitors: KML-29 (5 mg/kg), PF-3845 (3 mg/kg) and JZL-195 (20 mg/kg), or vehicle for the control group, were administered ip, 15 min before an acute ip injection of MK-801 (0.6 mg/kg) or vehicle. The mice were re-tested 15 min after the last injection (Table 2C).

### 4.6. Statistical Analysis

The statistical analysis was performed using one-way or two-way analysis of variance (ANOVA), for the factors of pre-treatment, treatment and pretreatment/treatment interactions. A post hoc comparison of means was carried out with Tukey’s test (for one-way ANOVA) or with Bonferroni’s test (for two-way ANOVA) for multiple comparisons, when appropriate.

The data were considered statistically significant at a confidence limit of *p* < 0.05. ANOVA with Tukey’s and Bonferroni’s post hoc tests was performed using GraphPad Prism version 7 for Windows, GraphPad Software, SanDiego CA, USA, www.graphpad.com; accessed on 8 July 2023.

For the memory-related behaviors, the changes in PA performance were expressed as the difference between retention and training latencies and were taken as the LI calculated for each animal and reported as the ratio mentioned before.

## 5. Conclusions

The influence on the memory and learning processes by the indirect influence of the ECS system (via FAAH and MAGL inhibitors) is controversial. There is literature data describing both positive and negative effects of various inhibitors on memory processes as well as memory disturbances associated with the dysfunction of other systems, e.g., the glutamatergic one, which is typical for pathogenesis of schizophrenia. However, based on the preclinical studies describing the positive impact on memory, the selected inhibitors may lead to new strategies to treat memory disorders associated with many mental disorders, including schizophrenia. KML-29 and PF-3845 normalized cognitive issues in an acute NMDA receptor antagonist model of schizophrenia in mice. Thus, such pharmacotherapy with inhibitors of endocannabinoids’ enzymatic degradation that could better-restore the deficits in cognition in patients with schizophrenia would be helpful.

## Figures and Tables

**Figure 1 ijms-24-11400-f001:**
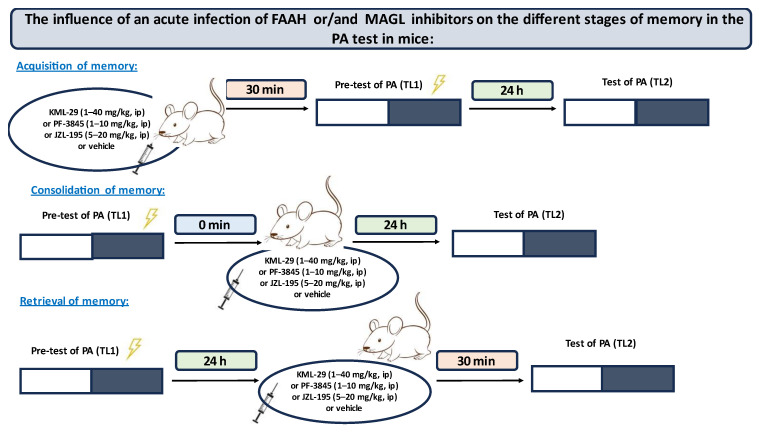
The scheme of the first step of the behavioral experiments.

**Figure 2 ijms-24-11400-f002:**
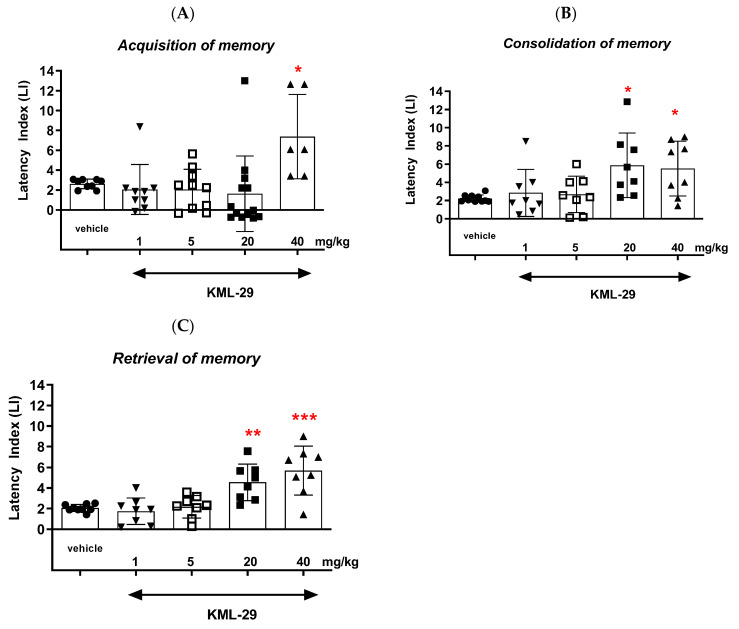
Effects of an acute administration of KML-29 on the memory-related responses expressed as the latency index (LI) during the acquisition (**A**), consolidation (**B**) and retrieval (**C**) memory using the PA test in mice. KML-29 (1–40 mg/kg) or vehicle for the control group were injected 30 min before the first trial (acquisition of memory), immediately after the first trial (consolidation of memory) or immediately before the second trial (retrieval of memory). Then, 24 h later, the second trial was conducted. n = 8–10, mean ± SEM; LI value for groups: ● vehicle; ▼ the dose of 1 mg/kg of KML-29; □ the dose of 5 mg/kg of KML-29; ◼ the dose of 20 mg/kg of KML-29; ▲ the dose of 40 mf/kg of KML-29; * *p* < 0.05, ** *p* < 0.01, and *** *p* < 0.001 vs. vehicle-treated group, Tukey’s test.

**Figure 3 ijms-24-11400-f003:**
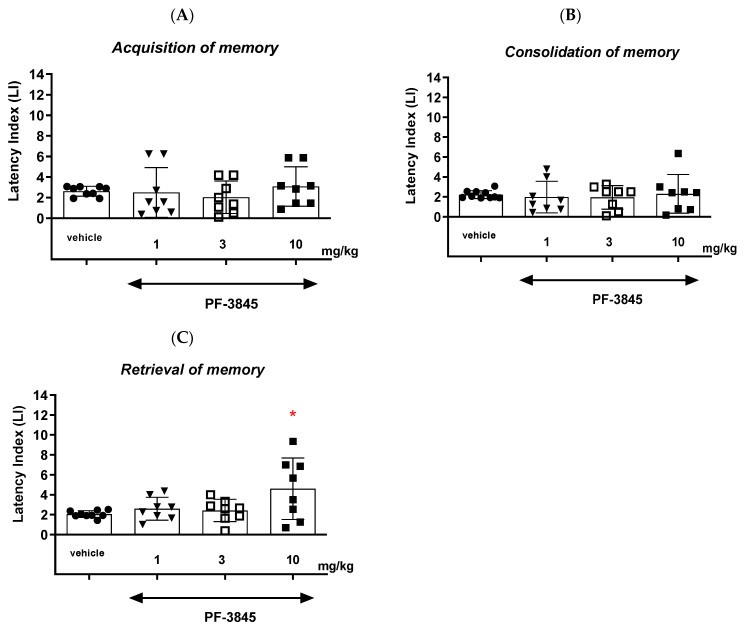
Effects of an acute administration of PF-3845 on the memory-related responses expressed as the latency index (LI) during the acquisition (**A**), consolidation (**B**) and retrieval (**C**) memory using the PA test in mice. PF-3845 (1–10 mg/kg) or vehicle for the control group were injected 30 min before the first trial (acquisition of memory), immediately after the first trial (consolidation of memory) or immediately before the second trial (retrieval of memory). Then, 24 h later, the second trial was conducted. n = 8–10, mean ± SEM; LI value for groups: ● vehicle; ▼ the dose of 1 mg/kg of PF-3845; □ the dose of 3 mg/kg of PF-3845; ◼ the dose of 10 mg/kg of PF-3845; * *p* < 0.05 vs. the vehicle-treated group, Tukey’s test.

**Figure 4 ijms-24-11400-f004:**
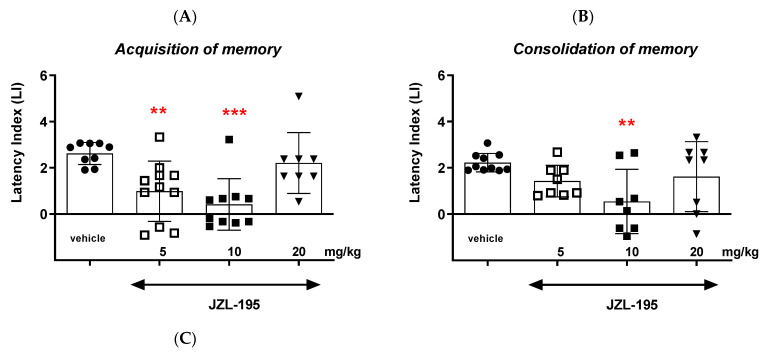
Effects of an acute administration of JZL-195 on the memory-related responses expressed as the latency index (LI) during the acquisition (**A**), consolidation (**B**) and retrieval (**C**) memory using the PA test in mice. JZL-195 (5–20 mg/kg) or vehicle for the control group were injected 30 min before the first trial (acquisition of memory), immediately after the first trial (consolidation of memory) or immediately before the second trial (retrieval of memory). Then, 24 h later, the second trial was conducted. N = 8–10, mean ± SEM; LI value for groups: ● vehicle; □ the dose of 5 mg/kg of JZL-195; ◼ the dose of 10 mg/kg of JZL-195; ▼ the dose of 20 mg/kg of JZL-195; * *p* < 0.05, ** *p* < 0.01, and *** *p* < 0.001 vs. the vehicle-treated group, Tukey’s test.

**Figure 5 ijms-24-11400-f005:**
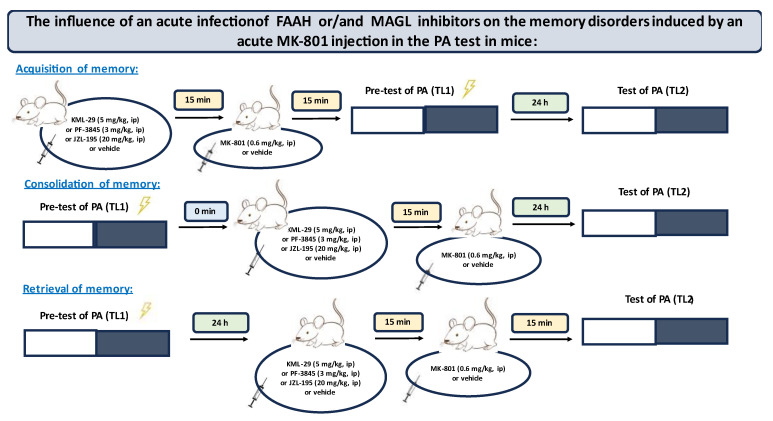
The scheme of the second step of the behavioral experiments.

**Figure 6 ijms-24-11400-f006:**
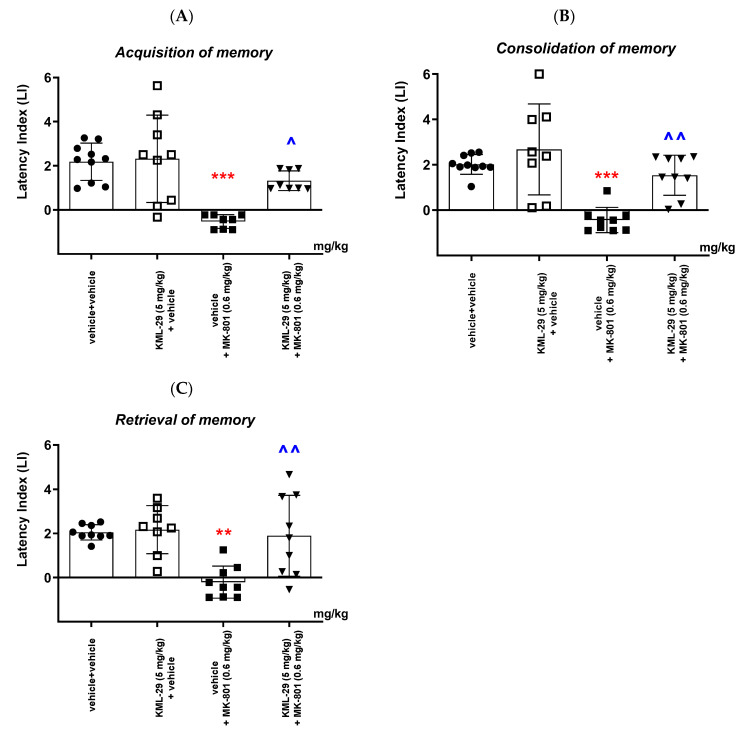
Effects of an acute administration of KML-29 (5 mg/kg) on the memory impairment provoked by MK-801 (0.6 mg/kg), expressed as the latency index (LI) during the acquisition (**A**), consolidation (**B**) and retrieval (**C**) memory using the PA test in mice. Non-effective doses of KML-29 (5 mg/kg) or vehicle were administered 15 min prior to the amnestic dose of MK-801 (0.6 mg/kg). These two injections were performed 15 before the first trial (acquisition of memory), immediately after the first trial (consolidation of memory) or immediately before the second trial (retrieval of memory). Then, 24 h later, the second trial was conducted. N = 8–10, mean ± SEM; LI value for groups: ● vehicle + vehicle; □ KML-29 (5 mg/kg) + vehicle; ◼ vehicle + MK-801 (0.6 mg/kg); ▼ KML-29 (5 mg/kg) + MK-801 (0.6 mg/kg); ** *p* < 0.01, *** *p* < 0.001 vs. the vehicle/vehicle group, ^ *p* < 0.05, ^^ *p* < 0.01 vs. the vehicle/MK-801 (0.6 mg/kg)-treated group, Bonferroni’s test.

**Figure 7 ijms-24-11400-f007:**
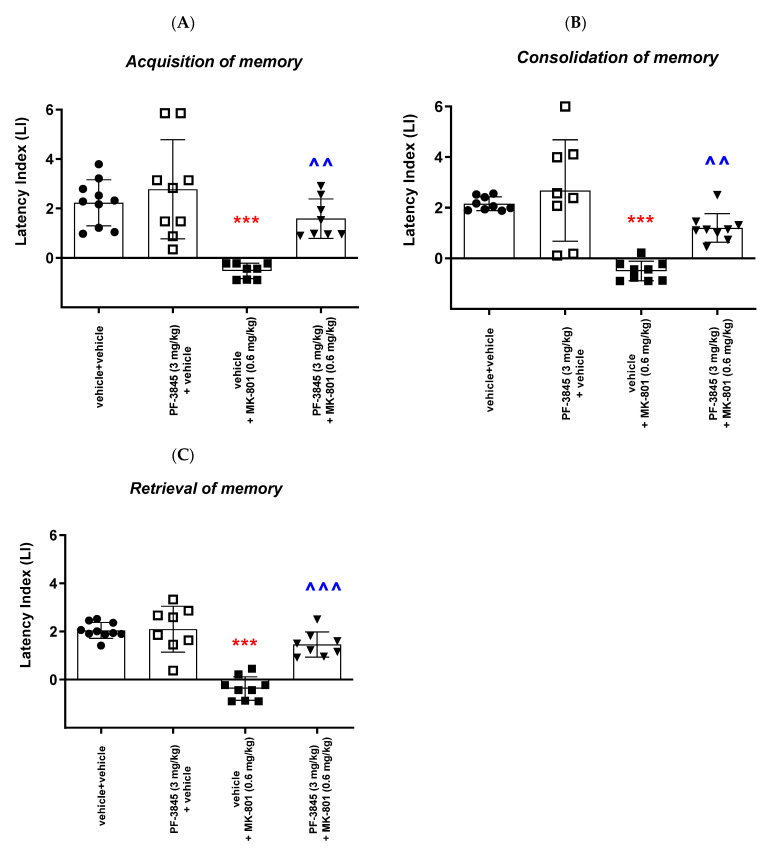
Effects of an acute administration of PF-3845 (3 mg/kg) on the memory impairment provoked by MK-801 (0.6 mg/kg), expressed as the latency index (LI) during the acquisition (**A**), consolidation (**B**) and retrieval (**C**) memory using the PA test in mice. Non-effective doses of PF-3845 (3 mg/kg) or vehicle were administered 15 min prior to the amnestic dose of MK-801 (0.6 mg/kg). These two injections were performed 15 before the first trial (acquisition of memory), immediately after the first trial (consolidation of memory) or immediately before the second trial (retrieval of memory). Then, 24 h later, the second trial was conducted. N = 8–10, mean ± SEM; LI value for groups: ● vehicle + vehicle; □ PF-3845 (3 mg/kg) + vehicle; ◼ vehicle + MK-801 (0.6 mg/kg); ▼ PF-3845 (3 mg/kg) + MK-801 (0.6 mg/kg); *** *p* < 0.001 vs. the vehicle/vehicle group, ^^ *p* < 0.01, ^^^ *p* < 0.001 vs. the vehicle/MK-801 (0.6 mg/kg)-treated group, Bonferroni’s test.

**Figure 8 ijms-24-11400-f008:**
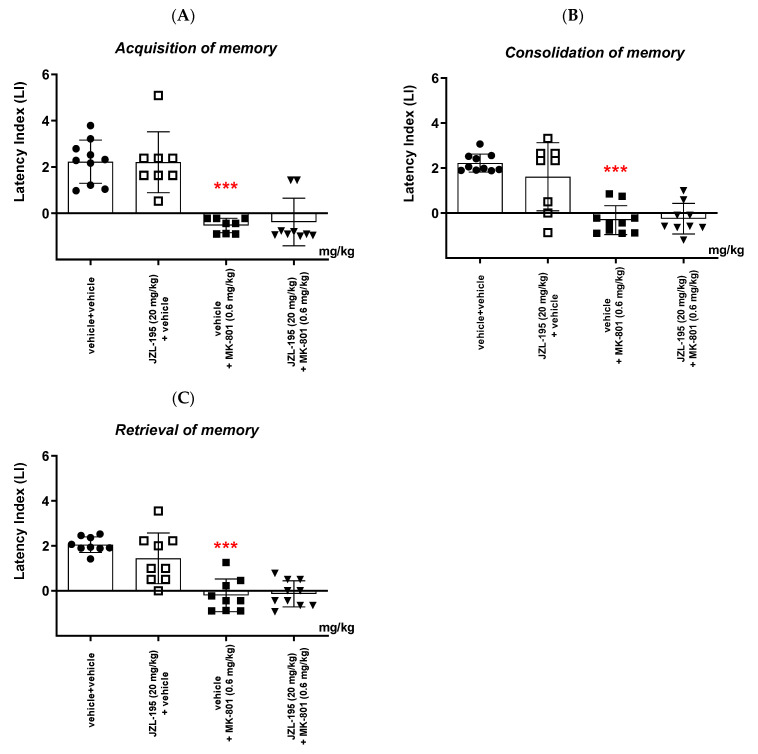
Effects of an acute administration of JZL-195 (20 mg/kg) on the memory impairment provoked by MK-801 (0.6 mg/kg), expressed as the latency index (LI) during the acquisition (**A**), consolidation (**B**) and retrieval (**C**) memory using the PA test in mice. Non-effective doses of JZL-195 (20 mg/kg) or vehicle were administered 15 min prior to the amnestic dose of MK-801 (0.6 mg/kg). These two injections were performed 15 min before the first trial (acquisition of memory), immediately after the first trial (consolidation of memory) or immediately before the second trial (retrieval of memory). Then, 24 h later, the second trial was conducted. n = 8–10, mean ± SEM; LI value for groups: ● vehicle + vehicle; □ JZL-195 (20 mg/kg) + vehicle; ◼ vehicle + MK-801 (0.6 mg/kg); ▼ JZL-195 (20 mg/kg) + MK-801 (0.6 mg/kg); *** *p* < 0.001 vs. vehicle/vehicle group, Bonferroni’s test.

**Figure 9 ijms-24-11400-f009:**
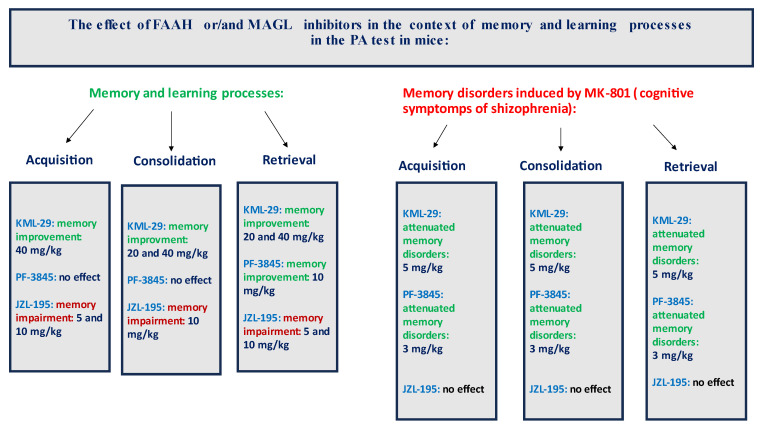
The graphical effects of all used inhibitors on the memory-related responses in mice in the PA test. Green font color—improvement of cognitive processes; red font color—impairment of cognitive processes; black font color—no influence on the cognitive processes.

**Table 1 ijms-24-11400-t001:** The scheme of inhibitors of enzymatic degradation of endocannabinoids or vehicle administration during the assessment of long-term memory acquisition (A), consolidation (B) or retrieval (C) in the PA test in mice.

(A)					
Acquisition of Memory					
PA Test	Drug Administration	Interval	TL1	Interval	TL2
Long-term memory	KML-29 (1, 5, 20, 40 mg/kg) PF-3845 (1, 3, 10 mg/kg) JZL-195 (5, 10, 20 mg/kg)	30 min	+	24 h	+
	vehicle	30 min	+	24 h	+
(**B**)					
**Consolidation of Memory**					
**PA Test**	**TL1**	**Interval**	**Drug Administration**	**Interval**	**TL2**
Long-term memory	+	0 min	KML-29 (1, 5, 20, 40 mg/kg) PF-3845 (1, 3, 10 mg/kg) JZL-195 (5, 10, 20 mg/kg)	24 h	+
	+	0 min	vehicle	24 h	+
(**C**)					
**Retrieval of Memory**					
**PA Test**	**TL1**	**Interval**	**Drug Administration**	**Interval**	**TL2**
Long-term memory	+	24 h	KML-29 (1, 5, 20, 40 mg/kg) PF-3845 (1, 3, 10 mg/kg) JZL-195 (5, 10, 20 mg/kg)	30 min	+
	+	24 h	vehicle	30 min	+

**Table 2 ijms-24-11400-t002:** The scheme of inhibitors of enzymatic degradation of endocannabinoids and MK-801 co-administration during the assessment of long-term memory acquisition (A), consolidation (B) or retrieval (C) in the PA test in mice.

(A)							
Acquisition of Memory							
PA Test	Drug Administration	Interval	Drug Administration	Interval	TL1	Interval	TL2
Long-term memory	KML-29 (5 mg/kg), PF-3845 (3 mg/kg), JZL-195 (20 mg/kg)	15 min	MK-801 (0.6 mg/kg) or vehicle	15 min	+	24 h	+
	vehicle (control group)	15 min	MK-801 (0.6 mg/kg) or vehicle	15 min	+	24 h	+
(**B**)							
**Consolidation of Memory**							
**PA Test**	**TL1**	**Interval**	**Drug Administration**	**Interval**	**Drug Administration**	**Interval**	**TL2**
Long-term memory	+	0 min	KML-29 (5 mg/kg), PF-3845 (3 mg/kg), JZL-195 (20 mg/kg)	15 min	MK-801 (0.6 mg/kg) or vehicle	24 h	+
	+	0 min	vehicle	15 min	MK-801 (0.6 mg/kg) or vehicle	24 h	+
(**C**)							
**Retrieval of Memory**							
**PA Test**	**TL1**	**Interval**	**Drug Administration**	**Interval**	**Drug Administration**	**Interval**	**TL2**
Long-term memory	+	24 h	KML-29 (5 mg/kg), PF-3845 (3 mg/kg), JZL-195 (20 mg/kg)	15 min	MK-801 (0.6 mg/kg) or vehicle	15 min	+
	+	24 h	vehicle	15 min	MK-801 (0.6 mg/kg) or vehicle	15 min	+

## Data Availability

Not applicable.

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
