# Peer review of "Inhibitors of Endocannabinoids’ Enzymatic Degradation as a Potential Target of the Memory Disturbances in an Acute N-Methyl-D-Aspartate (NMDA) Receptor Hypofunction Model of Schizophrenia in Mice"

_ijms, 2023, doi:10.3390/ijms241411400_

Round 1
Reviewer 1 Report
I am impressed with the writing of this manuscript. It covers many references in the indirect modulation of the endocannabinoid system for memory disorders.
The manuscript compared well-known FAAH and/or MAGL inhibitors in the same NMDA hypofunction animal model of schizophrenia.
However,
The scientific significance and novelty are moderate since all molecules tested in the manuscript have been tested in similar assay settings.
A clinical antipsychotic drug should be used as a positive control to validate the animal assay and the efficacy of these enzyme inhibitors.
The English language is well-written and scientific.
Author Response
Answer for Reviewer 1:
Thank you very much for the helpful suggestion. I agree with the Reviewer that the clinical antipsychotic drug should be used as a positive control to validate the animal assay and the efficacy of enzyme inhibitors. We are in the process of such research right now. The next publication, which is just being completed, concerns the assessment of the effect of combined administration of antipsychotics (aripiprazole and olanzapine) and selected inhibitors of endocannabinoid degradation in the brain in the context of modulation of cognitive disorders in an animal model of schizophrenia.
All changes made in the manuscript are marked in red font.

Reviewer 2 Report
The manuscript by Kruk-Slomka Marta et. al., titled “Inhibitors of endocannabinoids enzymatic degradation as a po-tential target of the memory disturbances in an acute N-methyl-D-aspartate (NMDA) receptor hypofunction model of schizo-phrenia in mice” is an interesting study that can be considered for publication with Major edits.
Major:
1. All the Figures should be represented as dot plot (shows complete data distribution) and mention number of animal used and number of animals responded
2. Give a graphical representation of effect of all inhibitors used linked to the result
3. In figure, please provide a time line of drug administration and test conducted
4. Limitations of the study need to be discussed in discussion
Minor
1. Figure 1 All the images should be kept in the same page (increase the X and Y axis’s fonts and reduce the size to fit all the panel in one page)
2. In the line 5 of discussion please correct the typo (sings is used instead of signs)
Author Response
Answer for Reviewer 2:
Major:
- As the Reviewer suggested all the Figures were changed as dot plot that show complete data distribution and number of animal used and number of animals responded. Additionally, the description of the number of animals used is placed below the Figures. Of course, if the Reviewer deems it necessary, we can add tables with the number of mice used and those that responded to the final version of the manuscript.
- As the Reviewer suggested a graphical representation of effect of all inhibitors used lined to the result should be added. The results obtained for all inhibitors are presented in detail in 7 graphs (Fig. 2-8). The graphs are made in the GraphPad Prism 8 program and clicking on the graph gives an additional opportunity to view individual results and the method of statistical analysis. Additionally the summary graph to show the effect of all inhibitors has been added at the end of Results Section (Fig. 9).
- The reviewer's suggestion was that the data on the experimental conditions (time line of drug administration and test conducted) should be included in the Figures. We tried to put such markings on the figures, however, due to the lack of readability of the graphs, we decided to create two separate Figures. The scheme of behavioral experiments (time line of drug administration and test conducted) was added before Figures in the Result Section (Fig. 1 and Fig. 5).
- The paragraph concerning the limitation of the study has been added at the end of Discussion Section.
Minor:
- As suggested by the Reviewer, the size of individual graphs was reduced so that each of the Figures fit on one page. However, the final version and appearance of the manuscript depends on the editors.
- In the Discussion Section the spelling error has been corrected.
All changes made in the manuscript are marked in red font.

Reviewer 3 Report
The manuscript presents an interesting research. The methods used seem to be correctly chosen and the conclusions reflect the results. The structure of the paper is a little difficult to follow. The authors present several section that repeat overall the same idea. I think the results could be presented in a more condensed manner. The figures could also be edited to be presented in a 2 x 2 square.
The discussion section is long and also hard to really follow. The authors could shortly describe the MAGL and FAAH inhibitors in the introduction section. See as example:
Zanfirescu et al. Targeting Monoacylglycerol Lipase in Pursuit of Therapies for Neurological and Neurodegenerative Diseases. Molecules. 2021;26(18):5668.
Targeting Endocannabinoid Signaling: FAAH and MAG Lipase Inhibitors, Annual Review of Pharmacology and Toxicology, Vol. 61:441-463, 2021.
In the discussion section I think it is important for the authors to mention that they consider that KML-29, PF-3845, and JZL-195 have only effects on their declared targets.
The overall editing of the paper should be corrected and simplified.
English is OK
Author Response
Answer for Reviewer 3:
- As the Reviewer suggested, the sentence that the inhibitors (KML-29, PF-3845, and JZL-195) were used for the assessment their influence on the memory-related effects and they have only effects on their declared targets has been added at the end of the Discussion section when discussing the limitations of the experiments.
- As the Reviewer suggested the manuscript has been corrected and simplified: discussion section has been corrected and shortened; a short description of FAAH and MAGL inhibitors has been moved to the Introduction section.
- As the Reviewer suggested all figures have been presented in a 2 x 2 square.
- In the review the Reviewer indicated that the “structure of the manuscript is a little difficult to follow”. Unfortunately, we cannot change the structure and layout of the manuscript as it is predetermined by editorial requirements.
All changes made in the manuscript are marked in red font.

Round 2
Reviewer 1 Report
The authors have addressed all comments. The manuscript can be accepted as the current form.
Reviewer 2 Report
I have no more suggestion and comments, as authors have answered our comments in this manuscript. This manuscript can be accepted in the present form.
Reviewer 3 Report
The authors improved the quality of their work.
English is OK